# Effect of Intake of Leucine-Rich Protein Supplement in Parallel with Resistance Exercise on the Body Composition and Function of Healthy Adults

**DOI:** 10.3390/nu14214501

**Published:** 2022-10-26

**Authors:** Gyu Seok Oh, Ju-hak Lee, Kyunghee Byun, Dong-Il Kim, Ki Deok Park

**Affiliations:** 1Department of Rehabilitation Medicine, Gil Medical Center, Incheon 21565, Korea; 2Department of Human Movement Science, Incheon National University, Incheon 21999, Korea; 3Functional Cellular Networks Laboratory, Lee Gil Ya Cancer and Diabetes Institute, Gachon University, Incheon 13120, Korea; 4Division of Health and Kinesiology, Incheon National University, Incheon 21999, Korea; 5Sports Functional Disability Institute, Incheon National University, Incheon 21999, Korea; 6Department of Rehabilitation Medicine, Gil Medical Center, Gachon University College of Medicine, Incheon 21565, Korea

**Keywords:** sarcopenia, leucine, lean body mass, skeletal muscle mass, elderly

## Abstract

Although sarcopenia has been dealt with in several studies, the standardized guidelines for preventing sarcopenia resulting from increased life expectancy are still insufficient. Therefore, this study evaluated the effects of daily resistance exercise and the intake of leucine-rich protein supplements daily for 12 weeks on the body composition and physical function of healthy adults aged >50 years living in Korea. The study analyzed 50 healthy people without medical conditions, who were randomly assigned to two groups (taking either protein powder or placebo powder) twice a day for 12 weeks. All participants performed resistance exercises regularly that could be repeated 8–12 times using a TheraBand for 12 weeks. A total of 41 participants completed the study. When measured via bioimpedance analysis (BIA), body fat mass (kg) and body fat (%) significantly decreased, and lean body mass (LBM) (kg) and skeletal muscle mass (SMM) (kg) significantly increased, in both groups. However, when measured via dual-energy X-ray absorptiometry (DXA), LBM was significantly increased only in the protein powder group. The LBM and SMM change measured via BIA was significantly greater in the protein powder group than in the placebo powder group (LBM: 0.95 ± 0.91 kg in the protein powder group vs. 0.38 ± 1.06 kg in the placebo powder group, *p* = 0.043; SMM: 0.69 ± 0.58 kg in the protein powder group vs. 0.29 ± 0.65 kg in the placebo powder group, *p* = 0.039, respectively). In the senior fitness test (SFT), significant functional improvement was found within the two groups, but no significant difference was found between the groups in the degree of improvement. In conclusion, in older people aged >50, to prevent sarcopenia, is more effective to combine resistance exercise and leucine-rich protein supplementation than to simply perform resistance exercise.

## 1. Introduction

The weakening of the musculoskeletal system and increasing fat mass are the most common body changes arising from aging. According to data from the National Statistical Office of the Republic of Korea, in 2020, people aged ≥65 years account for 15.7% of the population in Korea, and this is predicted to increase in the future, reaching 20.3% in 2025 and causing Korea to become a super-aged society.

According to previous studies, muscle mass and bone density decrease through various physiological changes with increasing age in the older population, and sarcopenia has been understood to be a natural phenomenon [1]. In addition, the aging problem and rapidly increasing economic medical costs are becoming a big topic worldwide [2]. However, recently, sarcopenia has been associated with decreased activity and increased fat mass, caused by increased insulin resistance and decreased physical endurance rather than simply increasing age [3].

Specifically, the loss of muscle mass in older people is associated with inadequate nutrition and nutritional deficiencies such as a lack of essential amino acids [4]. In a study reported in 2008, among people aged >70 years, only 40% appropriately ingested more than the recommended protein intake (0.8 g/kg/day), and in older people who consumed 1.1 g/kg/day of protein, the decrease in muscle mass was significantly less than that in older people who consumed 0.7–0.9 g/kg/day of protein [5].

In addition, several studies confirmed a significant decrease in muscle mass in older people whose protein intake was less than the recommended amount. Thus, maintaining protein intake above the recommended amount to compensate for this is necessary [6,7].

Common treatments for sarcopenia include resistance exercise to improve muscle mass and body function, and adequate protein and vitamin D intake [1]. For the absorption of proteins into the body and their appropriate use as people age, both the amount of protein consumed and the composition and quality of the protein ingested have an important effect [8]. Protein synthesis in skeletal muscle decreases with age, but is preserved when essential amino acids are ingested in more than the appropriate dose [9]. Specifically, leucine is one of the essential amino acids that cannot be synthesized in the body and can only be obtained through meat, fish, beans, or nuts. Leucine helps in skeletal muscle synthesis and protein balance by inducing the initiation of protein synthesis through an amino acid-sensing mechanism [10,11].

For effective protein synthesis in older people, consuming at least 2.5–2.8 g of leucine per meal may help prevent sarcopenia [12]. However, the results of the long-term use of leucine are still unclear. One study reported significant improvements in skeletal muscle mass in older people taking leucine-rich protein supplements for 3–4 months, but no muscle strength or functional changes were observed [13]. On the contrary, in a study of women living in Japan, no significant change was noted in skeletal muscle mass, but improvements in functions such as walking speed was noted [14]. In another study of healthy adults taking leucine-rich protein supplements for more than 8–16 months, skeletal muscle mass increased by 13–17% in areas including the arms and legs, and muscle strength was also improved [15]. In a study of older patients with type 2 diabetes, no significant changes were found in muscle function, body composition, or insulin sensitivity when consuming 2.5 g of leucine for ≥6 months [16].

Sarcopenia is caused by not only protein intake, but also increased muscle protein degradation and decreased muscle protein synthesis with age. Specifically, the decrease in activity increases resistance to stimulation for myofibrillar protein synthesis in the muscle [17]. In healthy older people, when the number of steps per day was reduced by approximately 76% for 2 weeks, insulin resistance increased by 12%, postprandial insulin sensitivity decreased by 43%, and myofibrillar protein synthesis decreased by 26% [18]. On the contrary, proper resistance exercise improves not only muscle strength but also muscular endurance, and positively affects body composition by affecting body fat reduction [19]. In other studies, resistance exercise, when regularly performed, had a positive effect on the musculoskeletal system, slowing the decrease in bone density or increasing bone density [20]. When resistance exercise was performed with different weights for 8 weeks, resistance exercise with a high weight that could be repeated 2–4 times was more effective in improving muscle mass than resistance exercise with a low weight that could be repeated 8–12 times [21]. Resistance exercise, with gradually increasing intensity, increases muscle mass and improves muscle strength in healthy older people [22].

Specifically, resistance exercises using elastic rubber bands, such as a TheraBand, can be freely adjusted according to the muscle strength or physical performance of the participants, so there is little risk of injury [23]. 

Several studies have reported the advantages of resistance exercise, and as life expectancy has recently increased, regular exercise is an essential factor for older people to maintain functional physical performance and muscle strength. However, to supplement protein consumed through exercise or muscle mass reduction caused by aging, additional appropriate protein supplementation is important [24]. When regular exercise and protein intake are combined, the timing of protein intake is also one of the important factors in muscle protein synthesis. The increase in the muscle cross-sectional area of type lla and type llx was more effective in the group taking protein immediately before and after exercise than in the group taking additional protein at breakfast and dinner time [20]. In another study, when the same exercise was performed, the group that consumed protein immediately after exercise showed more muscle mass improvement than the group that consumed protein 2 h after exercise [25].

Despite several studies on sarcopenia, the standardized guidelines for preventing sarcopenia due to increased life expectancy are still insufficient. Specifically, Asian people are skinnier than Western people and have a high possibility of significant physical decline even with a small amount of muscle loss [26]. In this regard, some studies in Korea have examined the effect of taking leucine-rich protein supplements on body composition [27], but no study has explored changes in body composition and body function when regular leucine-rich protein intake and simple daily resistance exercises are combined. In addition, suggesting the effect of protein is considered insufficient; this is because no previous studies have compared changes in simple resistance exercise in the same environment or protein intake rich in essential amino acids, such as leucine, in parallel with resistance exercise. Effective protein intake, as well as regular daily exercise, is also an important factor in preventing sarcopenia. In this study, resistance exercise was undertaken by all subjects and the amount of exercise was controlled, and to confirm the effect of leucine on muscle mass synthesis, body composition and functional improvement were measured. We hypothesize that sarcopenia prevention will be more effective if regular resistance exercise is combined with leucine-rich protein, which is important for protein synthesis through an amino acid-sensing mechanism. Additionally, if the elderly take supplements available on the market that contain enough leucine, they will be able to manage their nutrition more economically and efficiently.

Therefore, this study aimed to evaluate the effects of daily resistance exercise combined with daily intake of accessible leucine-rich protein supplements for 12 weeks on the body composition and physical function of healthy adults aged >50 years residing in Korea. Moreover, we would like to review the usefulness of leucine-rich protein supplementation in parallel with regular resistance exercise as a means to prevent sarcopenia.

## 2. Materials and Methods

### 2.1. Study Population

The study participants were Koreans aged 50–70 years living in Korea, and were recruited from November 2021 to February 2022 through a recruitment notice on the bulletin board of Gil Hospital in Incheon and at Incheon University Sports Center. 

The study participants consisted of healthy people who had no medical conditions and had no experience with drugs, including vitamin D and steroids, which could affect muscle strength, within 6 months of the start of the study. In addition, those who had regular exercise experience of at least 20 min at least 2 times per week within 6 months of the study were excluded. Among the participants who met these research conditions, 50 people voluntarily participated after hearing detailed explanations of the purpose and method of the study. 

### 2.2. Randomization and Blinding

After the baseline evaluation, participants were randomly assigned (1:1) to two groups taking either protein powder or placebo powder twice a day for 12 weeks. Randomization of intervention (protein powder) group and control (placebo powder) group at a 1:1 ratio was performed using the Research Randomizer software (version 4.0). Participants, evaluators, and researchers were blinded to the treatment allocation, maintaining a double-blind study trial design.

### 2.3. Blood Analysis

Blood collection was performed before the start of the study to determine whether the participants were clinically suitable for the study. With the consent of the participants, after maintaining an empty stomach for a total of ≥8 h, the effect of the diet was minimized, and blood collection was performed after stabilization. Blood was collected from the upper arm vein using a disposable syringe. A total of 1 mL of the collected blood was dispensed into ethylenediamine tetra-acetic acid tubes (EDTA tube, BD Vacutainer, Auckland, New Zealand), and then, a complete blood count and hemoglobin A1c were measured using a hemocytometer (XN-V, Sysmex, Tokyo, Japan). A total of 3 mL of the collected blood was aliquoted into a serum-separating tube (SST tube, BD Vacutainer, New Zealand) and centrifuged at 3000 rpm for 10 min. The separated serum was analyzed for aspartate aminotransferase, alanine aminotransferase, blood urea nitrogen, low-density lipoprotein, and triglyceride levels using an automatic blood biochemistry analyzer (7180, HITACHI, Tokyo, Japan).

### 2.4. Intervention

All study participants performed resistance exercises regularly for 12 weeks. This exercise program is based on the physical activity recommendations for elderly provided by The American College of Sports Medicine (ACSM) and progressively increases the load according to the individual’s level [28]. Resistance exercises were performed as a whole-body exercise, including the lower extremities and upper extremities, and were regularly conducted three times a week [29]. All resistance exercises were performed after sufficient practice time on postures and methods, under the guidance of an exercise expert, for each exercise session. The exercise time was a total of 60 min, including 10 min of warm-up, 30 min of resistive exercise, 10 min of circulating exercise, and 10 min of regular exercise. Until the first 4 weeks, resistance exercises using the body alone were performed, and thereafter, a total of 3 sets of resistance exercises at a level that could be repeated 8–12 times were performed using a TheraBand [30,31] (Table 1). The exercise intensity was changed every 4 weeks in the form of progressive resistance exercise and was evaluated using Borg’s rating of perceived exertion (RPE) scale [32]. Protein powder (Himmune Protein Balance, Ildong Foodis Co., Ltd., Seoul, Korea; carbohydrate: 12 g, fat: 3 g, protein: 20 g (total leucine 2000 mg), fructo-oligosaccharides: 3 g, calcium: 300 mg, magnesium: 150 mg, vitamin D: 400 IU (10 µg), vitamin B6: 3 mg, zinc: 9 mg, pantothenic acid: 5 mg) or placebo powder (carbohydrate: 38 g) was consumed twice a day with 150–200 mL of water, within 30 min before and after exercise on exercise days and within 30 min after breakfast and lunch on non-exercise days. Both protein powder and placebo powder are milky-white, matte powders that are odorless and have a mild sweet taste. In addition, the intake of separate protein supplements was prohibited, and unusual diets, drugs, and injections were controlled through a 24 h recall method. Regarding the subjects’ dietary intake, they maintained their original daily personal diet from before the study, which was managed by the 24 h recall method and food frequency questionnaire. 

### 2.5. Assessment

#### 2.5.1. Primary Outcome: Body Composition and Muscle Mass

Body composition was measured using bioimpedance analysis (BIA) and dual-energy X-ray absorptiometry (DXA). Inbody-720 was used for BIA analysis. All participants urinated before measurement and maintained a fasting state. Body fat mass (kg), body fat (%), fat-free mass (FFM; kg), lean body mass (LBM; kg), and skeletal muscle mass (SMM; kg) were measured using BIA, and LBM (kg), body fat mass (kg), body fat (%), and bone mineral density (g/cm^2^) were measured using DXA. For LBM (kg) and SMM (kg), the effect of height was corrected by dividing by the square of height. Measurements were performed twice, before and after the start of the study.

#### 2.5.2. Secondary Outcome: Muscle Strength 

Using a dynamometer (TAKEI, TKK 5401 Dynamometer, Tokyo, Japan), grip strength (kg) was measured twice in the right and left hands, and the maximum value was recorded. A push-up test was performed by supporting the knee on the ground and bending the elbow <90°, whereas the waist, hip, and thigh were in a straight line; this tect was performed to measure upper-body muscle strength. The maximum number of times that this could be performed without a time limit was measured. A plank exercise test was performed to evaluate core and abdominal strength. In the prone position, the wrist and elbow were placed on the ground, the arms and shoulders were fixed at right angles, and the sustainable time was measured by applying strength to the abdomen and hips and making the upper body and legs in a straight line.

#### 2.5.3. Secondary Outcome: Physical Performance

The senior fitness test (SFT) was performed to measure the physical performance of the participants before and after the start of the study [33]. The test consists of six items that can evaluate upper and lower extremity strength, muscular endurance, balance, flexibility, and agility (Table 2).

### 2.6. Statistical Analysis

The Kolmogorov–Smirnov and Shapiro–Wilk tests were used to confirm the normal distribution of the data collected in this study. The baseline clinical and biochemical parameters of the intervention and control groups were compared using the Mann–Whitney U test. The Mann–Whitney U test was also performed to compare the change between the two groups over 12 weeks. The Wilcoxon signed-rank test was performed to analyze changes between baseline and after 12 weeks in each group. A *p*-value of <0.05 was considered significant. The data in this study were analyzed using IBM SPSS Statistics for Windows, version 25.0 (IBM Corp., Armonk, NY, USA).

## 3. Results

### 3.1. Baseline Characteristics of the Participants

A total of 41 participants completed the study, which included 21 participants in the intervention group and 20 participants in the control group (Figure 1). A total of nine participants were dropped from the study: six due to withdrawal of consent and three were lost upon follow-up. No significant differences were found between the control group and the intervention group in the height, weight, waist circumference, blood pressure, body composition, muscle mass, muscle strength, and physical performance values evaluated before the start of the study (Table 3). In the blood analysis performed before the start of the study, no participant was judged clinically unsuitable for the study, and no significant differences were found between the control group and the intervention group in any of the items (Table 4). There was no significant change that was considered a side effect in the follow-up blood test performed after 12 weeks of leucine-rich protein intake in the intervention group (Table 5).

### 3.2. Changes within the Control Group during the 12-Week Study Period

Table 6 shows the changes in the control group combined with combined resistance exercise and placebo powder intake for 12 weeks. Weight (kg) decreased significantly at 12 weeks compared with that at the baseline (from 63.64 ± 12.26 to 62.67 ± 11.68) (Z = −2.296, *p* = 0.022). BMI (kg/m^2^) decreased significantly at 12 weeks compared with that at the baseline (from 24.00 ± 3.09 to 23.57 ± 2.94) (Z = −2.515, *p* = 0.012). Body fat mass (kg) measured via BIA decreased significantly at 12 weeks compared with that at the baseline (from 19.10 ± 4.58 to 17.81 ± 4.38) (Z = −3.436, *p* = 0.001). Body fat (%) measured via BIA decreased significantly at 12 weeks compared with that at the baseline (from 30.17 ± 5.37 to 28.54 ± 5.50) (Z = −3.585, *p* < 0.001). LBM (kg) measured via BIA increased significantly at 12 weeks compared with that at the baseline (from 41.97 ± 9.31 to 42.35 ± 9.21) (Z = −2.113, *p* = 0.035). The lean body mass index (LBMI; kg/m^2^) measured via BIA increased significantly at 12 weeks compared with that at the baseline (from 15.76 ± 2.30 to 15.90 ± 2.23) (Z = −2.128, *p* = 0.033). SMM (kg) measured via BIA increased significantly at 12 weeks compared with that at baseline (from 24.46 ± 5.97 to 24.75 ± 5.94) (Z = −2.218, *p* = 0.027). The skeletal muscle mass index (SMMI) (kg/m^2^) measured via BIA increased significantly at 12 weeks compared with that at the baseline (from 9.17 ± 1.54 to 9.28 ± 1.50) (Z = −2.133, *p* = 0.033). However, no significant change was found in body composition measured via DXA. The number of push-up reps increased significantly at 12 weeks compared with that at the baseline (from 17.95 ± 15.74 to 35.90 ± 16.12) (Z = −3.921, *p* < 0.001). In the SFT, significant changes were confirmed in four of the six items. The number of chair stand reps/30 s increased significantly at 12 weeks compared with that at the baseline (from 16.85 ± 5.10 to 23.55 ± 4.99) (Z = −3.779, *p* < 0.001). The number of biceps curl reps/30 s increased significantly at 12 weeks compared with that at the baseline (from 18.10 ± 6.11 to 23.40 ± 3.93) (Z = −3.303, *p* = 0.001). The chair sit-and-reach test (cm) improved significantly at 12 weeks compared with that at the baseline (from 8.96 ± 9.40 to 12.83 ± 9.00, Z = −2.704, *p* = 0.007). The number of steps/2 min in the 2 min step test increased significantly at 12 weeks compared with that at the baseline (from 109.75 ± 17.44 to 137.85 ± 16.98) (Z = −3.920, *p* < 0.001).

### 3.3. Changes within the Intervention Group during the 12-Week Study Period

Table 7 shows the changes in the intervention group with combined resistance exercise and protein powder intake for 12 weeks. No significant changes in body weight and BMI were found in the intervention group. Waist circumference (cm) decreased significantly at 12 weeks compared with the baseline (from 79.52 ± 9.96 to 77.59 ± 9.78) (Z = −2.883, *p* = 0.004). Body fat mass (kg) measured via BIA decreased significantly at 12 weeks compared with that at the baseline (from 16.95 ± 4.86 to 15.90 ± 4.84) (Z = −3.494, *p* < 0.001). Body fat (%) measured via BIA decreased significantly at 12 weeks compared with that at the baseline (from 27.48 ± 6.28 to 25.80 ± 6.43) (Z = −3.669, *p* < 0.001). FFM (kg) measured via BIA that did not change in the control group and increased significantly at 12 weeks compared with that at the baseline (from 44.69 ± 10.16 to 45.74 ± 10.63) (Z = −3.423, *p* = 0.001). LBM (kg) measured via BIA increased significantly at 12 weeks compared with that at the baseline (from 42.17 ± 9.61 to 43.12 ± 9.95) (Z = −3.436, *p* = 0.001). The LBMI (kg/m^2^) measured via BIA increased significantly at 12 weeks compared with that at the baseline (from 15.87 ± 2.22 to 16.22 ± 2.28) (Z = −3.397, *p* = 0.001). SMM (kg) measured via BIA increased significantly at 12 weeks compared with that at the baseline (from 24.62 ± 6.17 to 25.31 ± 6.42) (Z = −3.576, *p* < 0.001). The SMMI (kg/m^2^) measured via BIA increased significantly at 12 weeks compared with that at the baseline (from 9.25 ± 1.49 to 9.51 ± 1.54) (Z = −3.572, *p* < 0.001). Unlike the control group, a significant change in body composition measured via DXA was also confirmed in the intervention group. LBM (kg) measured via DXA increased significantly at 12 weeks compared with that at the baseline (from 41.57 ± 9.82 to 42.58 ± 10.33) (Z = −3.389, *p* = 0.001). The LBMI (kg/m^2^) measured via DXA increased significantly at 12 weeks compared with that at the baseline (from 15.64 ± 2.37 to 16.01 ± 2.48) (Z = −3.389, *p* = 0.001). Body fat (%) measured via DXA decreased significantly at 12 weeks compared with that at the baseline (from 29.76 ± 6.07 to 29.20 ± 5.98 (Z = −2.051, *p* = 0.040). Hand grip strength (kg), which had no change in the control group, increased significantly at 12 weeks compared with that at the baseline (from 28.67 ± 9.05 to 30.50 ± 9.56) (Z = −2.190, *p* = 0.029). The number of push-up reps increased significantly at 12 weeks compared with that at the baseline (from 18.86 ± 14.14 to 38.10 ± 15.70) (Z = −4.022, *p* < 0.001). The duration of the plank test (s), which did not change in the control group, improved significantly at 12 weeks compared with that at the baseline (from 125.90 ± 67.44 to 177.90 ± 76.54) (Z = −3.442, *p* = 0.001). Like the control group, significant changes were confirmed in four of the six items in the SFT. The number of reps/30 s in the chair stand test increased significantly at 12 weeks compared with that at the baseline (from 18.67 ± 4.76 to 26.43 ± 4.15) (Z = −4.025, *p* < 0.001). The number of reps/30 s in the biceps curl test increased significantly at 12 weeks compared with that at the baseline (from 22.00 ± 7.20 to 28.10 ± 4.97) (Z = −3.359, *p* = 0.001). The results of the chair sit-and-reach test (cm) improved significantly at 12 weeks compared with those at the baseline (from 9.67 ± 11.21 to 13.17 ± 10.86) (Z = −2.660, *p* = 0.008). The number of steps/2 min in the 2 min step test increased significantly at 12 weeks compared with that at the baseline (from 110.90 ± 17.39 to 145.71 ± 28.37) (Z = −3.859, *p* < 0.001).

### 3.4. Comparison of Changes over 12 Weeks in the Control Group and Intervention Group

Table 8 shows a comparison of the change in 12 weeks between the control group and the intervention group. The weight (kg) loss in 12 weeks was significantly greater in the control group than in the intervention group (−0.06 ± 1.29 in the intervention group vs. −0.97 ± 2.07 in the control group) (Z = −1.972, *p* = 0.049). The BMI (kg/m^2^) reduction in 12 weeks was significantly greater in the control group than in the intervention group (0.00 ± 0.42 in the intervention group vs. −0.43 ± 0.77 in the control group) (Z = −2.212, *p* = 0.027). In the comparison of changes in body composition measured via BIA, the improvement in indicators related to muscle mass was significantly higher in the intervention group than in the control group. The FFM (kg) increase measured via BIA in 12 weeks was significantly greater in the intervention group than in the control group (1.06 ± 1.00 in the intervention group vs. 0.37 ± 1.10 in the control group) (Z = −2.194, *p* = 0.028). The LBM (kg) increase measured via BIA in 12 weeks was significantly greater in the intervention group than in the control group (0.95 ± 0.91 in the intervention group vs. 0.38 ± 1.06 in the control group) (Z = −2.025, *p* = 0.043). The LBMI (kg/m^2^) increase measured via BIA in 12 weeks was significantly greater in the intervention group than in the control group (0.35 ± 0.33 in the intervention group vs. 0.14 ± 0.38 in the control group) (Z = −2.087, *p* = 0.037). The SMM (kg) increase measured via BIA in 12 weeks was significantly greater in the intervention group than in the control group (0.69 ± 0.58 in the intervention group vs. 0.29 ± 0.65 in the control group) (Z = −2.066, *p* = 0.039). The SMMI (kg/m^2^) increase measured via BIA in 12 weeks was significantly greater in the intervention group than in the control group (0.26 ± 0.21 in the intervention group vs. 0.11 ± 0.24 in the control group) (Z = −2.153, *p* = 0.031). Although the comparison of body composition changes measured via DXA was not significant, LBM (kg) and LBMI (kg/m^2^) changes tended to be larger in the intervention group than in the control group and were close to the significance level (LBM: 1.01 ± 1.16 in the intervention group vs. 0.18 ± 1.16 in the control group, *p* = 0.055; LBMI: 0.37 ± 0.40 in the intervention group vs. 0.06 ± 0.44 in the control group, *p* = 0.050, respectively). The plank test (s) improvement was significantly greater in the intervention group than in the control group (52.00 ± 50.37 in the intervention group vs. 5.40 ± 63.36 in the control group) (Z = −2.530, *p* = 0.011). In the comparison of the change in body function measured via SFT, no significant difference was found between the two groups.

## 4. Discussion

This study investigated the effect of leucine-rich protein supplementation in parallel with regular resistance exercise on the body composition and function of healthy adults aged >50 years living in Korea, and consequently, we may utilize it for the prevention of sarcopenia. Many studies have focused on protein intake and resistance exercise in older people [20,24,25], but no study has focused on the effect of exercising regularly and consuming leucine-rich protein on the body function and body composition of older people. Therefore, this study is intended to evaluate the effect of additional leucine-rich protein intake on body function and body composition during resistance exercise in the elderly and considers it an effective method that has a positive effect on the prevention of sarcopenia.

As people age, efforts to prevent various diseases and maintain and improve muscle mass are important for a better life. The decrease in muscle mass is caused by decreased physical activity and metabolic decline due to age increase, and this change is also seen in older people with constant body weight and BMI [34]. In addition, in previous studies of older people, physical strength was low not only in the group with excessively high BMI, but also in the group with a low BMI [35]. In a study conducted on Koreans aged >65 years, if there was weight loss, there was a high risk of functional deterioration, evaluated via daily living performance and instrumental daily living performance [36]. In older people, weight loss has various causes and mechanisms, and older people complaining of weight loss may suffer from total malnutrition, which may lead to cachexia [37]. Even if it is not cachexia, when there is weight loss related to aging, fat-free mass is also lost, which results in muscle loss and, consequently, functional decline [38]. Therefore, it is necessary to prevent muscle mass loss in older people and maintain body weight to reduce the risk of functional deterioration caused by aging.

Skeletal muscles are tissues that are highly adaptable to various stimuli and grow in response to nutrient or mechanical loads [39]. Resistance exercise causes skeletal muscle injury, increases immune reactants for homeostasis, and stimulates the phosphoinositide 3 kinase (PI3K)-protein kinase B (PKB/Akt)-mammalian target of rapamycin (mTOR) signaling pathway for protein synthesis [40].

In addition to resistance training, nutritional support can have a positive effect on muscle mass increase and muscle strength improvement. Among various nutrients, protein supplementation was found to positively affect muscle strength increase and prevent muscle mass loss [41]. Leucine, one of the amino acids that make up proteins, is the only dietary amino acid that stimulates the mTOR signaling pathway and is known to positively affect muscle mass growth by increasing muscle protein synthesis and reducing muscle protein degradation [42]. Therefore, it is important to maintain muscle mass not only through regular daily resistance exercise but also by consuming appropriate proteins needed for muscle synthesis.

In this study, a decrease in fat mass and an increase in muscle mass measured via BIA were confirmed in the control group with resistance exercise without supplementation of leucine-rich protein; however, no significance was confirmed for DXA, a more precise test, and a significant decrease in weight and BMI was observed (Table 5). In the intervention group with adequate intake of leucine-rich protein along with resistance exercise, body weight and BMI remained at the same level after 12 weeks. In the intervention group, for the decrease in fat mass measured via BIA, FFM and muscle mass significantly increased, and LBM measured via DXA also increased significantly (Table 6). The weight and BMI reductions were significantly greater in the control group than in the intervention group. On the contrary, the FFM increase and muscle mass increase measured via BIA were greater in the intervention group than in the control group. The average LBM increase measured via DXA was higher in the intervention group than in the control group and showed very close significance levels (Table 7). As a result, it was more effective to increase muscle mass without losing weight when resistance exercise was performed with adequate intake of a leucine-rich supplement than to simply perform resistance exercise. Unlike the expected results, no significant difference was found between the two groups in the degree of functional improvement measured via SFT.

Although Asian studies have combined resistance exercise and protein intake, it was not possible to accurately control protein intake by applying chicken breast or confirm the effect of essential amino acids such as leucine using protein supplements with unclear components [43]. In a study on Asians who consumed a leucine-rich supplement, it was not possible to control the exercise of the intake group and the non-intake group during the study period. Thus, it was considered insufficient to present the effect of a protein supplement rich in leucine [27]. On the contrary, this study controlled all groups to perform the same level of resistant exercise, as well as the timing and amount of leucine-rich protein intake; thus, leucine-rich protein intake may have possible effects.

Leucine is known to increase intracellular and extracellular signal transduction substances for increasing muscle mass, and to increase muscle protein synthesis through mTOR [38]. This study confirmed that the intake of leucine had positive effects on muscle mass increase and core muscle endurance improvement. As hypothesized before the start of the study, when resistance exercise that stimulates the mTOR signaling pathway was regularly performed, leucine had a synergistic effect on increasing muscle mass. In addition, the consumption of leucine is believed to have positive effects on body composition, such as a decrease in body fat and an increase in muscle mass while maintaining body weight, thereby having positive effects on the long-term function of older people.

Although not confirmed in this study, several studies have revealed biomarkers related to muscle synthesis. In addition to malnutrition and decreased physical activity, chronic inflammation and hormonal changes affect the incidence of sarcopenia in older people [44,45]. Specifically, aging is associated with changes in the production and sensitivity of insulin-like growth factor 1 (IGF-1), cortisol, testosterone, and estrogen. The decrease in IGF-1 in older people was found to be associated with an increase in visceral fat, a decrease in muscle mass, a decrease in bone density, and a decrease in muscle strength [46,47]. When body fat increases, elevated free fatty acids decrease growth hormone production and plasma IGF-1 concentration [48]. By contrast, resistance exercise activates IGF-1 and, consequently, has positive effects on protein synthesis and muscle fiber recovery [49]. During intense exercise, oxygen intake increases in skeletal muscles, most of which are used in the mitochondria, but a small amount of oxygen is converted to superoxide radicals, hydrogen peroxide, and hydroxyl radicals, which are classified as reactive oxygen species (ROS) [50]. Activated oxygen associated with oxidative stress in aging skeletal muscles is one of the main causes of sarcopenia, and in preparation for ROS-induced oxidative damage, skeletal muscle cells use antioxidants produced in the mitochondria such as superoxide dismutase (SOD), glutathione peroxidase (GPx), and catalase [51]. While studies have shown that SOD [52] and GPx [53] activities were increased in skeletal muscles during endurance exercises, some studies have reported no change in SOD [54] and GPx [55] activities due to training. At the cellular level of aging muscles, decreased muscle use acts as stress, activating the skeletal muscle atrophy mechanism, which is known as the main skeletal muscle atrophy factor and a signal carrier that breaks down muscle proteins [56]. On the contrary, in healthy adults, blood glucocorticoid levels were maintained low through IGF-1 signaling, and skeletal muscle atrophy was prevented by reducing the expression of MuRF-1 [57]. However, a follow-up study is needed to more clearly identify the effect of leucine treatment on protein synthesis mechanisms by simultaneously identifying biomarkers related to muscle synthesis.

This study has some limitations. First, this study included a relatively small sample size and was conducted in a single community-based center in Korea, which may limit the generalizability of the findings. In addition, because of the small sample size, we were unable to identify gender differences. In the future, follow-up studies involving a larger number of participants from centers in various regions are needed and it is necessary to check the differences according to gender. Second, since the participants were healthy adults without problems in their daily living, it was difficult to determine whether function was adequately improved for daily living. A study including patients receiving rehabilitation treatment at a rehabilitation center can determine whether functional recovery improves. Third, although changes in body composition were confirmed in this study, changes at the cellular level, such as mTOR signaling pathway activation, were not confirmed. Fourth, the protein supplement used in this study contains not only leucine but also other micronutrients such as calcium and vitamin D, which may cause bias that may affect the results. Lastly, the subjects’ usual leucine intake before the start of the study and additional leucine intake through a diet other than protein supplements during the study period were not checked.

Despite these limitations, this study has several strengths. By strictly controlling resistance exercise via trained experts, the effect of leucine-rich protein on body composition when combined with exercise was confirmed. In addition, during the study period, the diets of the participants were carefully monitored by an experienced dietitian, and other protein supplements, unusual diets, and drugs were controlled. Protein intake was strictly controlled using commercially available quantified protein supplements instead of using a non-quantified protein source. In this study, 40 g of protein containing 4000 mg of leucine was consumed per day for 12 weeks, and we confirmed that no clinical side effects were noted on follow-up blood tests. Therefore, the results of this study can be easily and efficiently applied to healthy adults in real life. Lastly, to confirm body composition, accuracy was increased using DXA as well as BIA.

## 5. Conclusions

In this study, we investigated the effect of additional leucine-rich protein supplement intake on muscle mass improvement during resistance exercise in healthy adults aged >50 years. When resistance exercise was controlled, the intake of leucine-rich protein supplements for 12 weeks helped improve muscle mass without weight loss. As suggested by the hypothesis, leucine-rich proteins appear to have a positive effect on muscle mass synthesis through the mTOR pathway during resistance exercise. We conclude that older people who engage in exercise should have proper intake of leucine-rich protein supplements, which are effective in preventing weight loss and sarcopenia and help improve muscle mass. Further carefully designed studies with larger sample sizes and longer follow-up periods are needed to confirm changes at the cellular level, such as the mTOR signaling pathway.

## Figures and Tables

**Figure 1 nutrients-14-04501-f001:**
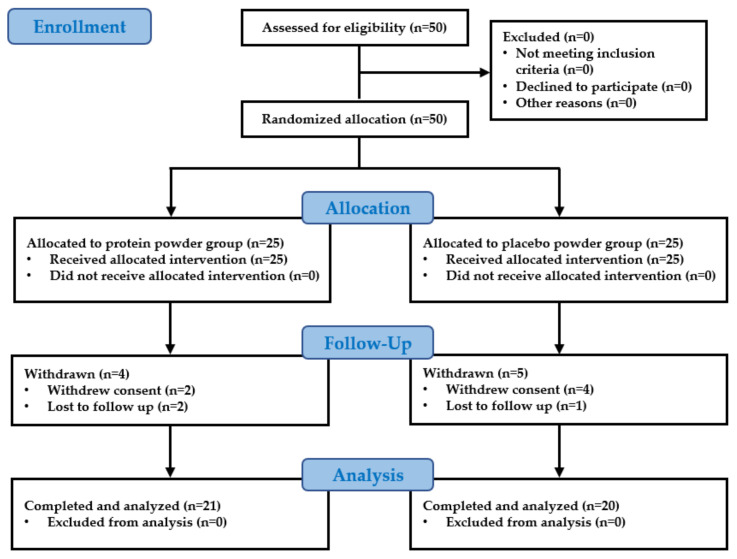
Flow diagram for the study population.

**Table 1 nutrients-14-04501-t001:** Exercise training program for 12 weeks (3 times/week).

Order	Time (min)	Contents	Duration (weeks)	Intensity
Warm-up	10′	Dynamic stretching	1–12	<RPE 5
Resistance exercise	30′	Body-weight training 1. Sit-to-stand 2. Body twist 3. Push-up against the wall 4. Sit and lift legs 5. Bridge exercise 6. Y raise 7. Knee extension 8. Back extension	1–4	RPE 5–6 8 reps each
		TheraBand exercises 1. Chest press 2. Bicep curl 3. Tricep extension 4. Side lateral raise 5. Front raise 6. Shoulder blade squeeze 7. Crunch 8. Trunk rotation 9. Squat 10. Leg press 11. Knee extension 12. Hip abduction	5–8	RPE 5–6 10 reps each
			9–12	RPE 5–6 12 reps each
Circuit training	10′	1. Chest press 2. Front raise 3. Squat 4. Crunch	1–12	RPE 7–8
Cool-down	10′	Static stretching	1–12	<RPE 5

RPE, rating of perceived exertion scale.

**Table 2 nutrients-14-04501-t002:** Senior fitness test.

Test Item	Test Description
Chair stand (number of stands)	Number of full stands in 30 s with arms folded across the chest
Arm curl (number of reps)	Number of bicep curls in 30 s holding a hand weight (men, 8 lb; women, 5 lb)
Chair sit-and-reach (cm +/−)	From a sitting position in front of a chair, with legs extended and hands reaching toward the toes, number of cm (+/−) from the extended fingers to the tip of the toes
Back scratch (cm +/−)	With one hand reaching over the shoulder and one up the middle of the back, number of cm between the extended middle fingers (+/−)
2.44 m up-and-go (s)	Number of seconds required to stand up from a seated position, walk 2.44 m, turn, and return to a seated position on a chair
2 min step (number of steps)	Number of full steps completed in 2 min, raising each knee to a point midway between the patella and iliac crest (number of times the right knee reaches the target)

**Table 3 nutrients-14-04501-t003:** Baseline characteristics of the participants.

	Mean ± S.E		Z	*p*
	Intervention (*n* = 21)	Control (*n* = 20)		
Height (m)	1.62 ± 0.08	1.62 ± 0.07	−0.065	0.948
Weight (kg)	61.65 ± 12.58	63.64 ± 12.26	−0.717	0.473
BMI (kg/m^2^)	23.26 ± 3.16	24.00 ± 3.09	−0.978	0.328
Waist circumference (cm)	79.52 ± 9.96	77.83 ± 17.95	−0.444	0.657
SBP (mmHg)	127.57 ± 18.38	125.55 ± 16.21	−0.235	0.814
DBP (mmHg)	83.19 ± 9.98	82.00 ± 11.40	−0.405	0.686
Body composition and muscle mass (Inbody-720)				
Body fat mass (kg)	16.95 ± 4.86	19.10 ± 4.58	−1.174	0.240
Body fat (%)	27.48 ± 6.28	30.17 ± 5.37	−1.448	0.148
Fat-free mass (kg)	44.69 ± 10.16	44.49 ± 9.79	−0.300	0.764
Lean body mass (kg)	42.17 ± 9.61	41.97 ± 9.31	−0.287	0.774
Lean body mass index (kg/m^2^)	15.87 ± 2.22	15.76 ± 2.30	−0.496	0.620
Skeletal muscle mass (kg)	24.62 ± 6.17	24.46 ± 5.97	−0.391	0.696
Skeletal muscle mass index (kg/m^2^)	9.25 ± 1.49	9.17 ± 1.54	−0.652	0.514
Body composition and muscle mass (DXA)				
Lean body mass (kg)	41.79 ± 9.15	41.57 ± 9.82	−0.156	0.876
Lean body mass index (kg/m^2^)	15.64 ± 2.37	15.71 ± 2.40	−0.052	0.958
Body fat mass (kg)	17.55 ± 4.51	19.28 ± 3.81	−0.991	0.322
Body fat (%)	29.76 ± 6.07	31.80 ± 4.23	−1.148	0.251
Bone mineral density (g/cm^2^)	1.17 ± 0.15	1.15 ± 0.10	−0.404	0.686
T-score	0.75 ± 1.38	0.54 ± 0.70	−0.183	0.855
Muscle strength				
Hand grip strength (kg)	28.67 ± 9.05	28.73 ± 10.17	−0.170	0.865
Push-ups (reps)	18.86 ± 14.14	17.95 ± 15.74	−0.379	0.705
Plank (s)	125.90 ± 67.44	135.45 ± 73.58	−0.235	0.814
Physical performance (senior fitness test)				
Chair stand test (reps/30 s)	18.67 ± 4.76	16.85 ± 5.10	−0.929	0.353
Bicep curl test (reps/30 s)	22.00 ± 7.20	18.10 ± 6.11	−1.672	0.095
2.44 m up-and-go test (s)	4.62 ± 0.68	4.76 ± 0.93	−0.196	0.845
Chair sit-and-reach test (cm)	9.67 ± 11.21	8.96 ± 9.40	−0.522	0.601
Back scratch test (cm)	−5.31 ± 11.34	−3.80 ± 10.48	−0.731	0.465
2 min step test (steps/2 min)	110.90 ± 17.39	109.75 ± 17.44	−0.575	0.565

BMI, body mass index; SBP, systolic blood pressure; DBP, diastolic blood pressure; DXA, dual-energy X-ray absorptiometry.

**Table 4 nutrients-14-04501-t004:** Baseline blood test of participants.

	Mean ± S.E		Z	*p*
	Intervention (*n* = 21)	Control (*n* = 20)		
Hb (g/dL)	13.30 ± 1.33	14.14 ± 1.26	−1.880	0.060
WBC (×10^3^ cells/μL)	5.82 ± 1.51	5.55 ± 1.80	−0.574	0.566
AST (U/L)	28.39 ± 11.01	29.82 ± 20.80	−0.013	0.990
ALT (U/L)	25.63 ± 17.68	24.36 ± 10.38	−0.130	0.896
HbA1c (%)	5.67 ± 0.45	5.77 ± 0.45	−0.694	0.487
BUN (mg/dL)	17.30 ± 4.55	17.09 ± 3.29	−0.143	0.886
LDL (mg/dL)	103.40 ± 27.57	100.42 ± 30.96	−0.469	0.639
TG (mg/dL)	129.71 ± 122.64	117.80 ± 72.70	−0.404	0.686

ALT, alanine aminotransferase; AST, aspartate aminotransferase; BUN, blood urea nitrogen; Hb, hemoglobin; HbA1c, hemoglobin A1c; LDL, low-density lipoprotein; TG, triglyceride; WBC, white blood cell count.

**Table 5 nutrients-14-04501-t005:** Blood test changes within the intervention group (*n* = 21) during the 12-week study period.

	Mean ± S.E		Z	*p*
	Baseline	12 Weeks		
Hb (g/dL)	13.30 ± 1.33	13.39 ± 1.23	−0.806	0.420
WBC (×10^3^ cells/μL)	5.82 ± 1.51	5.95 ± 1.56	−0.817	0.414
AST (U/L)	28.39 ± 11.01	26.65 ± 6.87	−0.504	0.614
ALT (U/L)	25.63 ± 17.68	24.49 ± 11.09	−0.278	0.781
HbA1c (%)	5.67 ± 0.45	5.60 ± 0.31	−0.088	0.930
BUN (mg/dL)	17.30 ± 4.55	19.39 ± 4.38	−1.808	0.071
LDL (mg/dL)	103.40 ± 27.57	108.60 ± 32.29	−1.356	0.175
TG (mg/dL)	129.71 ± 122.64	155.29 ± 186.00	−0.417	0.677

ALT, alanine aminotransferase; AST, aspartate aminotransferase; BUN, blood urea nitrogen; Hb, hemoglobin; HbA1c, hemoglobin A1c; LDL, low-density lipoprotein; TG, triglyceride; WBC, white blood cell count.

**Table 6 nutrients-14-04501-t006:** Changes within the control group (*n* = 20) during the 12-week study period.

	Mean ± S.E		Z	*p*
	Baseline	12 Weeks		
Weight (kg)	63.64 ± 12.26	62.67 ± 11.68	−2.296	0.022 *
BMI (kg/m^2^)	24.00 ± 3.09	23.57 ± 2.94	−2.515	0.012 *
Waist circumference (cm)	77.83 ± 17.95	80.46 ± 9.42	−1.457	0.145
Body composition and muscle mass (Inbody-720)				
Body fat mass (kg)	19.10 ± 4.58	17.81 ± 4.38	−3.436	0.001 **
Body fat (%)	30.17 ± 5.37	28.54 ± 5.50	−3.585	<0.001 ***
Fat-free mass (kg)	44.49 ± 9.79	44.86 ± 9.67	−1.945	0.052
Lean body mass (kg)	41.97 ± 9.31	42.35 ± 9.21	−2.113	0.035 *
Lean body mass index (kg/m^2^)	15.76 ± 2.30	15.90 ± 2.23	−2.128	0.033 *
Skeletal muscle mass (kg)	24.46 ± 5.97	24.75 ± 5.94	−2.218	0.027 *
Skeletal muscle mass index (kg/m^2^)	9.17 ± 1.54	9.28 ± 1.50	−2.133	0.033 *
Body composition and muscle mass (DXA)				
Lean body mass (kg)	41.57 ± 9.82	41.97 ± 9.27	−0.971	0.332
Lean body mass index (kg/m^2^)	15.71 ± 2.40	15.77 ± 2.43	−0.896	0.370
Body fat mass (kg)	19.28 ± 3.81	19.13 ± 3.37	−0.299	0.765
Body fat (%)	31.80 ± 4.23	31.64 ± 4.36	−1.090	0.276
Bone mineral density (g/cm^2^)	1.15 ± 0.10	1.14 ± 0.11	−1.139	0.255
T-score	0.54 ± 0.70	0.39 ± 1.12	−1.101	0.271
Muscle strength				
Hand grip strength (kg)	28.73 ± 10.17	29.79 ± 8.43	−1.248	0.212
Push-ups (reps)	17.95 ± 15.74	35.90 ± 16.12	−3.921	<0.001 ***
Plank (s)	135.45 ± 73.58	140.85 ± 64.41	−0.483	0.629
Physical performance (senior fitness test)				
Chair stand test (reps/30 s)	16.85 ± 5.10	23.55 ± 4.99	−3.779	<0.001 ***
Bicep curl test (reps/30 s)	18.10 ± 6.11	23.40 ± 3.93	−3.303	0.001 **
2.44 m up-and-go test (s)	4.76 ± 0.93	4.81 ± 0.78	−0.093	0.926
Chair sit-and-reach test (cm)	8.96 ± 9.40	12.83 ± 9.00	−2.704	0.007 **
Back scratch test (cm)	−3.80 ± 10.48	−4.15 ± 11.38	−0.078	0.938
2 min step test (steps/2 min)	109.75 ± 17.44	137.85 ± 16.98	−3.920	<0.001 ***

*** *p* < 0.001, ***p* < 0.01, * *p* < 0.05. BMI, body mass index; DXA, dual-energy X-ray absorptiometry.

**Table 7 nutrients-14-04501-t007:** Changes within the intervention group (*n* = 21) during the 12-week study period.

	Mean ± S.E		Z	*p*
	Baseline	12 Weeks		
Weight (kg)	61.65 ± 12.58	61.59 ± 12.80	−0.168	0.866
BMI (kg/m^2^)	23.26 ± 3.16	23.26 ± 3.14	−0.357	0.721
Waist circumference (cm)	79.52 ± 9.96	77.59 ± 9.78	−2.883	0.004 **
Body composition and muscle mass (Inbody-720)				
Body fat mass (kg)	16.95 ± 4.86	15.90 ± 4.84	−3.494	<0.001 ***
Body fat (%)	27.48 ± 6.28	25.80 ± 6.43	−3.669	<0.001 ***
Fat-free mass (kg)	44.69 ± 10.16	45.74 ± 10.63	−3.423	0.001 **
Lean body mass (kg)	42.17 ± 9.61	43.12 ± 9.95	−3.436	0.001 **
Lean body mass index (kg/m^2^)	15.87 ± 2.22	16.22 ± 2.28	−3.397	0.001 **
Skeletal muscle mass (kg)	24.62 ± 6.17	25.31 ± 6.42	−3.576	<0.001 ***
Skeletal muscle mass index (kg/m^2^)	9.25 ± 1.49	9.51 ± 1.54	−3.572	<0.001 ***
Body composition and muscle mass (DXA)				
Lean body mass (kg)	41.79 ± 9.15	42.58 ± 10.33	−3.389	0.001 **
Lean body mass index (kg/m^2^)	15.64 ± 2.37	16.01 ± 2.48	−3.389	0.001 **
Body fat mass (kg)	17.55 ± 4.51	17.39 ± 4.30	−0.678	0.498
Body fat (%)	29.76 ± 6.07	29.20 ± 5.98	−2.051	0.040 *
Bone mineral density (g/cm^2^)	1.17 ± 0.15	1.16 ± 0.14	−1.905	0.057
T-score	0.75 ± 1.38	0.91 ± 1.35	−1.170	0.242
Muscle strength				
Hand grip strength (kg)	28.67 ± 9.05	30.50 ± 9.56	−1.248	0.029 *
Push-ups (reps)	18.86 ± 14.14	38.10 ± 15.70	−3.921	<0.001 ***
Plank (s)	125.90 ± 67.44	177.90 ± 76.54	−3.442	0.001 **
Physical performance (senior fitness test)				
Chair stand test (reps/30 s)	18.67 ± 4.76	26.43 ± 4.15	−4.025	<0.001 ***
Bicep curl test (reps/30 s)	22.00 ± 7.20	28.10 ± 4.97	−3.359	0.001 **
2.44 m up-and-go test (s)	4.62 ± 0.68	4.36 ± 0.44	−0.956	0.339
Chair sit-and-reach test (cm)	9.67 ± 11.21	13.17 ± 10.86	−2.660	0.008 **
Back scratch test (cm)	−5.31 ± 11.34	−3.82 ± 10.29	−0.825	0.409
2 min step test (steps/2 min)	110.90 ± 17.39	145.71 ± 28.37	−3.859	<0.001 ***

*** *p* < 0.001, ** *p* < 0.01, * *p* < 0.05. BMI, body mass index; DXA, dual-energy X-ray absorptiometry.

**Table 8 nutrients-14-04501-t008:** Comparison of changes in 12 weeks in the control group and intervention group.

	Mean ± S.E		Z	*p*
	Intervention (*n* = 21)	Control (*n* = 20)		
Δ Weight (kg)	−0.06 ± 1.29	−0.97 ± 2.07	−1.972	0.049 *
Δ BMI (kg/m^2^)	0.00 ± 0.42	−0.43 ± 0.77	−2.212	0.027 *
Δ Waist circumference (cm)	−1.94 ± 3.12	2.64 ± 18.11	−0.575	0.565
Body composition and muscle mass (Inbody-720)				
Δ Body fat mass (kg)	−1.05 ± 0.84	−1.29 ± 1.63	−0.196	0.845
Δ Body fat (%)	−1.67 ± 1.14	−1.63 ± 1.83	−1.109	0.267
Δ Fat-free mass (kg)	1.06 ± 1.00	0.37 ± 1.10	−2.194	0.028 *
Δ Lean body mass (kg)	0.95 ± 0.91	0.38 ± 1.06	−2.025	0.043 *
Δ Lean body mass index (kg/m^2^)	0.35 ± 0.33	0.14 ± 0.38	−2.087	0.037 *
Δ Skeletal muscle mass (kg)	0.69 ± 0.58	0.29 ± 0.65	−2.066	0.039 *
Δ Skeletal muscle mass index (kg/m^2^)	0.26 ± 0.21	0.11 ± 0.24	−2.153	0.031 *
Body composition and muscle mass (DXA)				
Δ Lean body mass (kg)	1.01 ± 1.16	0.18 ± 1.16	−1.917	0.055
Δ Lean body mass index (kg/m^2^)	0.37 ± 0.40	0.06 ± 0.44	−1.956	0.050
Δ Body fat mass (kg)	−0.16 ± 0.82	−0.16 ± 1.56	−0.261	0.794
Δ Body fat (%)	−0.57 ± 1.12	−0.16 ± 2.00	−0.744	0.457
Δ Bone mineral density (g/cm^2^)	−0.01 ± 0.02	0.00 ± 0.02	−0.457	0.648
Δ T-score	0.16 ± 0.42	−0.15 ± 0.65	−1.526	0.127
Muscle strength				
Δ Hand grip strength (kg)	1.82 ± 3.43	1.07 ± 4.37	−0.457	0.648
Δ Push-ups (reps)	19.24 ± 8.98	17.95 ± 9.81	−0.405	0.685
Δ Plank (s)	52.00 ± 50.37	5.40 ± 63.36	−2.530	0.011 *
Physical performance (senior fitness test)				
Δ Chair stand test (reps/30 s)	7.76 ± 3.86	6.70 ± 4.54	−0.760	0.447
Δ Bicep curl test (reps/30 s)	6.10 ± 6.11	5.32 ± 5.40	−0.380	0.704
Δ 2.44 m up-and-go test (s)	−0.27 ± 0.81	0.04 ± 0.74	−0.626	0.531
Δ Chair sit-and-reach test (cm)	3.50 ± 6.47	3.87 ± 5.48	−0.118	0.906
Δ Back scratch test (cm)	1.49 ± 6.90	−0.35 ± 4.03	−0.707	0.480
Δ 2 min step test (steps/2 min)	34.81 ± 28.97	28.10 ± 16.90	−0.652	0.514

* *p* < 0.05.; Δx, x change after 12 weeks (x after 12 weeks—x at baseline); BMI, body mass index; DXA, dual-energy X-ray absorptiometry.

## Data Availability

The data that support the findings of this study are available from the corresponding author upon reasonable request.

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
