# Peer review of "Effect of Intake of Leucine-Rich Protein Supplement in Parallel with Resistance Exercise on the Body Composition and Function of Healthy Adults"

_nutrients, 2022, doi:10.3390/nu14214501_

Round 1

Reviewer 1 Report

The authors propose a very interesting application for the management of sarcopenia.

I think the two things have to go further, that is protein intake and physical activity with resistance opposition.

Some points to improve:

- The title should be revised a bit, it would seem that the supplement has added leucine, in fact, leucine is naturally present in the proteins used, in particular, whey.

- Why choice of a supplement with different protein sources, moreover of vegetable and animal origin?

- The supplement contains other micronutrients that are not present in the placebo, it would have been better if it had not, it should be emphasized as a possible bias

- It would have been interesting to have groups with just supplementation and exercise only

- Nothing is said about gender, are there any differences?

- I would like to comment on the organization of physical exercise which is in line with, for example, 10.3389 / fspor.2022.950949, perhaps to be mentioned as confirmation, however, the "seniority" of training of the subjects should be indicated and if a period was operated preliminary to practice in the exercises

Author Response

Response to Reviewer 1 Comments

Dear Editor,

We would like to express our heartfelt thanks for the review of our paper and insightful comments. Your remarks and suggestions have helped us revise our manuscript into a more valuable article and improve our research skills for future studies.  

We revised our manuscript according to the reviewers’ comments, as followings(Revisions in the manuscript are marked in red.):

Point 1: The title should be revised a bit, it would seem that the supplement has added leucine, in fact, leucine is naturally present in the proteins used, in particular, whey

Response 1: Thanks for your thoughtful comment. In this study, we considered leucine as a key ingredient in protein supplements, and further described hypotheses and mechanisms for the effects of leucine (Line 71-73, Line 127-130, Line 419-422, Line 454-456). Therefore, leucine is emphasized in the title.

Point 2: Why choice of a supplement with different protein sources, moreover of vegetable and animal origin?

Response 2: Thanks for your thoughtful comment. If a commercially available protein supplement with quantified content is used for the study, it is easy to keep the protein quantity constant during the study period. As a result, it is possible to reduce the bias caused by the irregular protein amount (Line 442-445, Line 505-507). And if we use a product that is easily available in the market, we thought that it could be applied economically and efficiently in the future (Line 507-510).

Point 3: The supplement contains other micronutrients that are not present in the placebo, it would have been better if it had not, it should be emphasized as a possible bias

Response 3: Thanks for your thoughtful comment. We added the possibility of bias due to other micronutrients (Line 498-500).

 Point 4: It would have been interesting to have groups with just supplementation and exercise only

Response 4: Thanks for your thoughtful comment. There have been previous studies in which only protein supplements were taken without controlling the amount of exercise (Line 445-448). And it would have been interesting to include groups that only took supplements or only exercised. However, in order to see the effect of the protein containing leucine with a small number of samples, resistance exercise was performed in common.

 Point 5: Nothing is said about gender, are there any differences?

Response 5: Thanks for your thoughtful comment. Due to the small sample size, gender differences could not be identified. It was additionally described as a limitation (Line 489-492).

 Point 6: I would like to comment on the organization of physical exercise which is in line with, for example, 10.3389 / fspor.2022.950949, perhaps to be mentioned as confirmation, however, the "seniority" of training of the subjects should be indicated and if a period was operated preliminary to practice in the exercises

Response 6: Thanks for your thoughtful comment. We have revised the content related to the organization of physical exercise (Line 173-184).

Reviewer 2 Report

The manuscript was prepared very well. However, there are some concerns, in part important, so the review articles need revision, see below.

Introduction

·       Include which components of leucine-rich protein supplement have the effects indicated to meet the objectives of the study

·       Include previous history of similar investigations and justify the need for this investigation

Materials and Methods

·       How was the randomization process carried out?

·       hs registered the diet of the participants with a FFQ?

·       accurately describe the composition and nature of the supplement used

·       indicate the methodological process according to the CONSORT rules

·       Has this study passed an ethics committee? Has it been carried out according to the Helsinki rules and the Fortalzeza declaration?

Results

·       The results should be presented in clearer tables, redo the rows and columns to improve the presentation of results

·       Include the Consort diagram in the development of the study

Discussion

·       You must include references in those paragraphs that do not contain them, so that its content is justified.

·       It should include some hypothesis of the possible mechanisms described from a physiological perspective or include some mechanism of action described.

·       It should include some comparative discussion with other studies related to its purpose, explaining the differences.

·       What does this study contribute? Clarify.

·       Any possible application of the results described?

·       Include a section on strengths and limitations.

Conclusion

·       In the Conclusion section, state the most important outcome of your work. Do not simply summarize the points already made in the body — instead, interpret your findings at a higher level of abstraction. Show whether, or to what extent, you have succeeded in addressing the need stated in the Introduction (or objectives).

Author Response

Response to Reviewer 2 Comments

Dear Editor,

We would like to express our heartfelt thanks for the review of our paper and insightful comments. Your remarks and suggestions have helped us revise our manuscript into a more valuable article and improve our research skills for future studies.  

We revised our manuscript according to the reviewers’ comments, as followings(Revisions in the manuscript are marked in red.):

Point 1: Include which components of leucine-rich protein supplement have the effects indicated to meet the objectives of the study.

 Response 1: Thanks for your thoughtful comment. In this study, we considered leucine as a key ingredient in protein supplements, and further described hypotheses and mechanisms for the effects of leucine (Line 71-73, Line 127-130, Line 419-422, Line 454-456).

Point 2: Include previous history of similar investigations and justify the need for this investigation

Response 2: Thanks for your thoughtful comment. A related past study was described (Line 74-83), and there was a study that took only protein rich in leucine, but it was difficult to see the additional effect of leucine because exercise was not controlled (Line 119-122). In this regard, the necessity of this study was further described (Line 127-130).

Point 3: How was the randomization process carried out?

Response 3: Thanks for your thoughtful comment. We have described the content related to the randomization process (Line 154-158).

 Point 4: has registered the diet of the participants with a FFQ?

Response 4: Thanks for your thoughtful comment. We have described the contents of the subject's diet management (Line 195-197).

 Point 5: accurately describe the composition and nature of the supplement used

Response 5: Thanks for your thoughtful comment. In addition to the components of the supplement used, the characteristics are further described(Line 186-190, Line 192-193)

 Point 6: indicate the methodological process according to the CONSORT rules

Response 6: Thanks for your thoughtful comment. In compliance with the CONSORT rules, study population recruitment (Line 141), randomization (Line 152), intervention (Line 172), and assessment (Line 202) were described in the order. The diagram was modified based on the CONSORT rule to be more clear(Figure 1).

 Point 7: Has this study passed an ethics committee? Has it been carried out according to the Helsinki rules and the Fortalzeza declaration?

Response 7: Thanks for your thoughtful comment. We have complied with the Helsinki rules and the Fortalzeza declaration and passed the ethics committee(Approval No. GFIRB2021-376) (Line 533-534).

 Point 8: The results should be presented in clearer tables, redo the rows and columns to improve the presentation of results

Response 8: Thanks for your thoughtful comment. As suggested, we revised tables to improve the presentation of results (Table 3, Table 4, Table 5, Table 6, Table 7, Table 8)

 Point 9: include the Consort diagram in the development of the study

Response 9: Thanks for your thoughtful comment. The diagram was modified based on the CONSORT rule to be more clear (Figure 1).

 Point 10: You must include references in those paragraphs that do not contain them, so that its content is justified

Response 10: Thanks for your thoughtful comment. We added references to the paragraphs where the reference is missing (Line 392).

 Point 11: It should include some hypothesis of the possible mechanisms described from a physiological perspective or include some mechanism of action described.

Response 11: Thanks for your thoughtful comment. In this study, we hypothesized that there would be a synergistic effect through the mTOR pathway when additional leucine-rich proteins were taken during resistance exercise (Line 412-415, Line 419-422, Line 454-456)

 Point 12: What does this study contribute? Clarify.

Response 12: Thanks for your thoughtful comment. We have more clearly clarified why we conducted our research and what we want to contribute(Line 388-390, Line 393-396).

Point 13: Any possible application of the results described?

Response 13: Thanks for your thoughtful comment. We confirmed that there was no side effect for 12 weeks through blood tests, and since we used supplements with quantified ingredients, we think that the research results can be applied in practice(Line 505-510).

Point 14: Include a section on strengths and limitations.

Response 14: Thanks for your thoughtful comment. As suggested, we have added a section on strengths and limitations(Line 487-511)

Point 15: In the Conclusion section, state the most important outcome of your work. Do not simply summarize the points already made in the body — instead, interpret your findings at a higher level of abstraction. Show whether, or to what extent, you have succeeded in addressing the need stated in the Introduction (or objectives).

Response 15: Thanks for your thoughtful comment. As suggested, we revised and reorganized the conclusion section (Line 513-523).

Round 2

Reviewer 1 Report

The authors made significant improvements to the manuscript, but some concern remains:

- About the choice of supplement, I could agree on having a commercial one, but I did not understand the choice of having three different protein sources; it is well known, for example, that the leucine-richest protein source is whey, so why do not have exclusively?  If the manufacturer has supported the project, it should be highlighted.

- In the weakness should be highlighted that the previous/actual leucine intake was not checked

Author Response

Dear Editor,

We would like to express our heartfelt thanks for the review of our paper and insightful comments. Your remarks and suggestions have helped us revise our manuscript into a more valuable article and improve our research skills for future studies.  

We revised our manuscript according to the reviewers’ comments, as followings(Revisions in the manuscript are marked in red.):

Point 1: About the choice of supplement, I could agree on having a commercial one, but I did not understand the choice of having three different protein sources; it is well known, for example, that the leucine-richest protein source is whey, so why do not have exclusively?  If the manufacturer has supported the project, it should be highlighted.

 Response 1: Thanks for your thoughtful comment. The product was funded for research. Considering the content of the supported products, we selected the supplement containing the most leucine. We have indicated that the product has been supported by the manufacturer(Line 532-534).

Point 2: In the weakness should be highlighted that the previous/actual leucine intake was not checked

Response 2: Thanks for your thoughtful comment. We added that we did not check previous leucine intake and the additional leucine intake by diet other than protein supplements during the study period(Line 500-502).

Reviewer 2 Report

the authors have assumed and completed the proposed changes. For my part there are no more suggestions

Author Response

Dear Editor,

We would like to express our heartfelt thanks for the review of our paper and insightful comments. Your remarks and suggestions have helped us revise our manuscript into a more valuable article and improve our research skills for future studies.  

We sincerely thank you again for your in-depth advice on our study.